# Loss in the Antibacterial Ability of a PyrR Gene Regulating Pyrimidine Biosynthesis after Using CRISPR/Cas9-Mediated Knockout for Metabolic Engineering in *Lactobacillus casei*

**DOI:** 10.3390/microorganisms11102371

**Published:** 2023-09-22

**Authors:** Shaojun Chen, Xinmiao He, Ziliang Qin, Gang Li, Wentao Wang, Zida Nai, Yaguang Tian, Di Liu, Xinpeng Jiang

**Affiliations:** 1College of Animal Science and Technology, Northeast Agricultural University, Harbin 150030, China; chen05160023@163.com (S.C.); qzlneau@163.com (Z.Q.); ligangneau@163.com (G.L.); tianyaguang2011@163.com (Y.T.); 2Key Laboratory of Combining Farming and Animal Husbandry, Ministry of Agriculture, Animal Husbandry Research Institute, Heilongjiang Academy of Agricultural Sciences, No. 368 Xuefu Road, Harbin 150086, China; haashxm@163.com (X.H.); wangwentao_1981@163.com (W.W.); 3College of Agriculture, Yanbian University, Yanji 133002, China; nzdneau@163.com

**Keywords:** CRISPR/Cas9, *Lactobacillus casei*, PyrR, pyrimidine biosynthesis

## Abstract

*Lactobacillus casei* (*L. casei*) has four possible mechanisms: antimicrobial antagonism, competitional adhesion, immunoregulation, and the inhibition of bacterial toxins. To delineate the metabolic reactions of nucleotides from *L. casei* that are associated with mechanisms of inhibiting pathogens and immunoregulation, we report that a PyrR-deficient *L. casei* strain was constructed using the CRISPR-Cas9^D10A^ tool. Furthermore, there were some changes in its basic biological characterization, such as its growth curve, auxotroph, and morphological damage. The metabolic profiles of the supernatant between the PyrR-deficient and wild strains revealed the regulation of the synthesis of genetic material and of certain targeting pathways and metabolites. In addition, the characteristics of the PyrR-deficient strain were significantly altered as it lost the ability to inhibit the growth of pathogens. Moreover, we identified PyrR-regulating pyrimidine biosynthesis, which further improved its internalization and colocalization with macrophages. Evidence shows that the PyrR gene is a key active component in *L. casei* supernatants for the regulation of pyrimidine biosynthesis against a wide range of pathogens.

## 1. Introduction

*Lactic acid bacteria* (LAB) are a heterogeneous group of Gram-positive bacteria with probiotic functions and a high tolerance for low pH. Live lactic acid bacteria have favorable effects on human and animal health. One study indicated that probiotics significantly reduced antibiotic-associated diarrhea by 52% and acute diarrhea of various causes by 34% [1]. Molecular and genetic studies have allowed for the determination of the basics of pathogen inhibition with probiotics via four mechanisms: antimicrobial antagonism from metabolic products, competition with pathogens for adhesion, immunoregulation [2], and the inhibition of bacterial toxins [3]. Various studies have indicated that LAB inhibit the growth of pathogenic microorganisms and degrade mycotoxins. LAB are abundantly found during the removal and degradation of different types of microorganisms, and they act by disrupting and changing the permeability of the plasma membrane, producing metabolites, and inhibiting protein translation. However, the optimum growth of LAB necessitates complex nutritional requirements for amino acids, peptides, nucleotide bases, vitamins, minerals, fatty acids, and carbohydrates. Meanwhile, LAB can also produce a variety of products, including short-chain fatty acids, amines, bacteriocins, vitamins, and exopolysaccharides, during its metabolism. The concrete antibacterial mechanisms of LAB are complicated, and this sense of complicatedness has been attributed to their ability to produce antimicrobial compounds, including bacteriocins, with strong competitive action against many microorganisms [4]. The proteomics of probiotics provide a special reference and tool through which we can analyze lactic acid bacteria to better understand their metabolic function in order to answer questions regarding basic cell functions. Nevertheless, single and multiple omics techniques may allow us to gain mechanistic insights and discover new metabolic properties with respect to LAB.

Genome editing tools can be used to overcome existing challenges in all of these perspectives of traditional, emerging, and future applications of lactic acid bacteria. The genome editing tools for LAB focus on a fast speed, clean and targeted modifications, and stable genomic genotypes for a wide variety of strains, which are crucial for the construction of strains for fundamental studies, applications, and products. Most metabolic reactions in LAB are connected via the utilization of nucleotides or in their metabolite regulation of nucleotides, such as in pyrimidine and purine metabolism. The detection of novel pyrimidine derivatives represents a promising area for new antibacterial drug discovery. These derivatives are the means by which one can interact with genetic materials, enzymes, and other biopolymer substances in the cell. Pyrimidine-containing agents represent major areas in the search for new antibacterial drugs due to their strong activities and diverse mechanisms, which represent an advantage in endogenous substances and in the derivatives that interact with genetic materials, enzymes, and other biopolymer substances in the cell [5]. The formation of pyrimidines has been found to occur in different steps in the biosynthetic pathway of LAB, as well as in the biosynthetic genes of pyrimidine that are scattered on the chromosome of the L. lactis subspecies for five transcriptional units [6]. The inactivation of PyrR in *Lactococcus lactis* may result in a three- to eight-fold de-repression in the expression of all of the pyrimidine biosynthetic genes that play an attenuating mechanistic role with the PyrR regulon [7]. Given that the PyrR from *B. subtilis* possesses significant enzymatic activity in vitro and in vivo [8], in previous studies, no research was conducted on metabolic regulation regarding the inactivation gene method in the PyrR of *lactobacillus*. The deletion of the PyrR gene via genome editing using CRISPR-CAS9 provided the possibility to assess the regulating activity of pyrimidine biosynthesis in LAB.

In this study, we used the systematic biochemical properties of CRISPR-Cas9^D10A^ in order to engineer it into an efficient genome editing tool for *Lactobacillus casei*. We combined rational engineering and constructed a PyrR-deficient *Lactobacillus casei* strain that is able to decrease the production of the pyrimidine analog, which also extensively interferes with the growth on different pathogenic bacteria. Moreover, we identified that the pyrimidine analog exhibits immunoregulatory activity in the PBMC. The aim of our study is to provide an overview of the physiology and antibacterial effects of nucleotide metabolism and its regulation, which will facilitate the interpretation of data arising from genetics, metabolomics, proteomics, and transcriptomics in *Lactobacillus casei*.

## 2. Results

### 2.1. Knockout of Pyrimidine Biosynthesis of PyrR Gene Using CRIPSR-Cas9^D10A^ Edition in L. casei

Based on the CRISPR-Cas9^D10A^ genome-editing plasmid pLCNICK, the pyrimidine regulatory gene (PyrR) was deleted in order to evaluate the knockout performance with the target of pyrimidine production. As is shown in the schematic diagram of the pLCNICK-mediated PyrR knockout, all the strains and constructing primers, as well as the plasmids from our studies, are shown in Table 1 and Table 2. The PyrR gene, as well as the upstream/downstream homology arms (HASup and HASdown), was cloned using PCR assays with a genome DNA template from *L. casei*. This was achieved by using the primer pyrR-ex-up/down, which was constructed as the plasmid of pMD19-PyrR, as shown in Table 1. The cloning vector carrying the PyrR and the upstream/downstream homology arm gene were cloned in order to conduct the upstream homology arm and downstream homology arm gene, respectively. These genes were connected in the cloning vector and were named pMD19-PyrRHAS-up and pMD19-PyrRHAS-down. Both the PyrRHAS-up and PyrRHAS-down genes were connected by the fusion PCR assays in which the PCR product was cloned into the plasmid of pMD19-PyrR-HAS. The sgRNA was cloned in front of the HAS in the plasmid of pMD19-PyrR-HAS-sgRNA, which was then ligated into the pLCNICK with the XbaI and ApaI restriction endonuclease sites. The genome-editing plasmid of PyrR was named pLCNICK-PyrR-HAS-sgRNA. This plasmid of pLCNICK-PyrR-HAS-sgRNA, containing Cas9, homologous arm (HAS) cassette genes, and SgRNA, targets the PyrR gene in *Lactobacillus* and guides the cleavage at the targeted gene and the HAS DNA, which can be inserted into the designed restriction sites. The recombinant nutritionally deficient strain was constructed via an electro-transformation in the *Lactobacillus casei*, which was grown in MRS medium plates supplemented with erythromycin for 4 days. The clones were selected to grow in the MRS medium with erythromycin resistance (Figure 1A). The genome PCR results from different clones indicated that PyrR was completely knocked out in *L. casei* ΔPyrR (Figure 1B). The PCR products were then sent to a gene sequencing company, and the sequence results indicated that the PyrR genes were deleted in the gene site. The *L. casei* and *L. casei* ΔPyrR samples were analyzed using Western blot bands, which showed that the *L. casei* ΔPyrR strain did not express the PyrR protein; however, it was found in the *L. casei* wild strain (Figure 1C).

### 2.2. Biological Characterization of L. casei ΔPyrR

To further characterize the deletion of the PyrR-gene-obtained *L. casei* and *L. casei* ΔPyrR mutants, the growth curves were analyzed, and their results showed that the *L. casei* and *L. casei* ΔPyrR were both S-shaped (Figure 2A). Both of them entered the logarithmic growth phase at the eighth hour, but the number of *L. casei* ΔPyrR was lower than in the wild strain of *L. casei*. They entered the stable phase at about the 16th–18th hour, at which point they then entered the decline phase. The difference was that the *L. casei* ΔPyrR directly entered the decline phase, meaning that it entered this phase earlier than the wild strain of *L. casei*. In addition, the bacterial count of *L. casei* ΔPyrR was always lower than in the wild strain of *L. casei* over the entirety of the growth period. Furthermore, pyrR encodes an mRNA-binding regulatory protein that negatively regulates pyrR expression by sensing UMP or UTP. The mutant strain of *L. casei* ΔPyrR was auxotrophic for uracil and 5-FOA resistance, whereas *L. casei* ΔPyrR was prototrophic for uracil (Figure 2B). The 5-FOA resistance analysis showed that the wild strain failed to grow well on the plates containing 5-FOA, whereas the *L. casei* ΔPyrR mutants grew well on the 5-FOA MRS plate. These results further confirm the PCR results, thus highlighting the feasibility of using the PyrR gene as a auxotrophic marker in *L. casei*.

### 2.3. Morphological Characteristics of Genome Editing in Lactobacillus casei

In order to explore whether the deletion of the PyrR gene affects growth characteristics, *L. casei* and *L. casei* ΔPyrR (in the logarithmic growth phase and stable phase) were subjected to scanning electron microscopy, and the morphological characteristics of the bacteria were thus observed (Figure 3). The scanning electron microscopy results also indicated that, compared with *L. casei*, the morphological characteristics of *L. casei* ΔPyrR were significantly changed. In both the 12th and 24th hours, the membranes of the cells remained integrated, and there was no leakage in the samples, which suggests that the genome editing treatment had no obvious impact on cell morphology. However, the cells became much shorter, and a serious leakage of the cell contents was observed after the genome editing treatment resulted in the deletion of the PyrR gene. Therefore, the deletion of the PyrR gene affected the enhancement of cell membrane permeability.

### 2.4. Changes in the Metabolic Profiles of L. casei after Editing the PyrR Gene

To identify the metabolites that are produced by lactobacilli deleting the PyrR gene and candidates for the microbiome-mediated regulation of pyrimidine synthesis, we assessed the metabolome with non-targeted metabolomics analysis to identify the bioactive small molecules in *L. casei* cell-free supernatants such as *L. casei* and *L. casei* ΔPyrR (Figure 4). A total of 1040 metabolites were detected in the supernatants of *L. casei* and *L. casei* ΔPyrR, and the major difference was found to be the existence of about 71 metabolites with positive-ion (53 metabolites, Figure 4A) and negative-ion (18 metabolites, Figure 4B) modes (which are depicted in Figure 4). When organized into chemical classes and metabolic pathways, the screening of major differences in the metabolites were classified into those associated with biological processes from the positive-ion (Figure 4D) and negative-ion (Figure 4C) modes. From these differentially abundant metabolites, the positive-ion-mode metabolites mainly included the following: (1) nucleosides, nucleotides, and analogs; (2) organic acids and derivatives; and (3) organoheterocyclic compounds. Additionally, the negative-ion-mode metabolites mainly included the following: (1) benzenoids; (2) organic acids and derivatives; and (3) organoheterocyclic compounds. The presence of pyrimidine, purine, and secondary metabolites of the antibacterial ingredients was significantly decreased in the *L. casei* ΔPyrR group (such as in uracil, virginiamycin, and L-(+)-lactic acid). Naturally, these differentially abundant metabolites were used in our metabolic pathway enrichment analysis. Based on the KEGG database, the significantly enriched metabolic pathways in the differential metabolites are shown in Figure 4E. Each bubble stands for one of the signaling pathways in the bacterial biosynthesis of *L. casei*, such as the biosynthesis of various secondary metabolites and pyridine alkaloid biosynthesis. All of these enriched metabolic pathways were closed by the PyrR gene in the regulation of the synthesis of genetic material. 

### 2.5. Interaction between Pathway and Metabolite Set Enrichment with the Addition of PyrR

The abundances of differential pathways based on the integrated metabolites were found by comparing *L. casei* ΔPyrR and *L. casei*, as shown in Figure 5A. The pathway analysis and metabolite set enrichment analysis were performed to generate a differential abundance score, which served to indicate the average and total changes in the metabolites that were enriched in a specific pathway, such as the cAMP signaling pathway; protein digestion and absorption; aminoacyl-tRNA biosynthesis; and tropane, piperidine, and pyridine alkaloid biosynthesis. Following a comparison of the different pathways, pyrimidine and purine metabolism (Figure 5B) and the biosynthesis of secondary metabolites (Figure 5C) were found to be the metabolic pathways most affected by the PyrR gene. The expression of differential metabolites in the pyrimidine and purine pathways showed that deoxyadenosine, uracil, D-ornithine, lactic acid, and citrate were down-regulated by the PyrR gene via the genome editing of *L. casei*, as is shown in Figure 5B. The biosynthesis of secondary metabolites was found to have occurred due to the enriched sources of drug leads and, in particular, in the numerous metabolites that were isolated from lactobacillus, as is shown in Figure 5C. PyrR gene deletion decreased the production of secondary metabolites in *L. casei* ΔPyrR when compared with *L. casei*. To explore the potential metabolic microbiota interactions between *L. casei* ΔPyrR and *L. casei*, a correlation network analysis was conducted using the differential metabolites, which was achieved by deleting the PyrR gene in lactobacillus (Figure 5D). The results of the correlation network demonstrate that there were complex connections between the differential metabolites and that the key metabolites experienced a reduction in deoxyadenosine, uracil, mefenamic acid, and pyrimidin-4-amine.

### 2.6. Antibacterial Activities following the Deletion of the PyrR Gene in L. casei

The antibacterial activities of the nucleic acid produced by *L. casei and L. casei* ΔPyrR were evaluated against *Bacillus subtilis*, *Staphylococcus aureus*, and *Salmonella* via the Oxford cup diffusion method, as shown in Table 2. The bacterial inhibition diameter (mm) reflected the antibacterial activity with the culture suspension from *L. casei and L. casei* ΔPyrR at different doses (i.e., 200 µL and 100 µL). In addition, PBS buffer and antibiotics were used as the negative and positive control groups, where a larger antibacterial diameter corresponded to a better antibacterial effect in the positive control group, the *L. casei* group, and the *L. casei* ΔPyrR group. As shown in Table 1, almost all of the culture suspensions from *L. casei* and *L. casei* ΔPyrR produced a large or small clear inhibition zone against *Bacillus subtilis*, *Staphylococcus aureus*, and *Salmonella*. From the results, we found that the *L. casei* suspension showed high sensitivity (i.e., between 15–20 mm) and moderate sensitivity (between 10–14 mm) in quantities of 200 µL of the three pathogenic bacteria (*Bacillus subtilis*, *Staphylococcus aureus*, and *Salmonella*). However, the nucleic acid deficiency in the *L. casei* ΔPyrR groups led to a small clear inhibition zone when compared with the wild strain group, which also exhibited moderate sensitivity to antibacterial activity in quantities of 200 µL. There was no change in the inhibition zone in the wild strain of *L. casei* against the three pathogenic bacteria in a dose-dependent manner. However, the *L. casei* ΔPyrR groups contained a significantly dose-dependent level of pathogenic bacteria when compared in quantities of 200 µL and 100 µL. The suspension was found to have a low sensitivity to *Staphylococcus aureus* with 100 µL of the *L. casei* ΔPyrR group. The best antibacterial activity was displayed against *Salmonella*, with high sensitivity. These results also displayed the same trend when compared with two further pathogenic bacteria. According to Table 3 all of the above results indicate that the antibacterial activity of *L. casei* was caused by the presence of pyrimidine metabolites in the suspension, not by the genome editing of *L. casei* ΔPyrR.

### 2.7. The Interaction between Peritoneal Macrophages and Genome Editing of L. casei in Regulating Cell Polarization

To ascertain the role of the deletion of the PyrR gene in the *L. casei*-associated pyrimidine metabolites, wild *L. casei* was used to stimulate the differentiation and polarization of the peritoneal macrophages in mice, which were treated with a co-culture system of macrophages that were separately extracted from other mice, as is shown in Figure 6. Both the peritoneal macrophages and *L. casei* were stained with F4/80+ and CFDA-SE markers, respectively. Compared to the control group without *L. casei*, our microbiome analysis revealed that both *L. casei* and *L. casei* ΔPyrR could dramatically increase the percentage of the activity of peritoneal macrophages in the 12th hour and 24th hour. Additionally, the twelfth-hour activity in the *L. casei* and *L. casei* ΔPyrR groups had a much better effect than that in the twenty-fourth hour when comparing these two groups. Moreover, the *L. casei* group induced obvious macrophage differentiation when compared with the *L. casei* ΔPyrR group in the 12th and 24th hours. Notably, there was no significant difference between the *L. casei* and *L. casei* ΔPyrR groups. However, the activity of the peritoneal macrophages in the 12th hour for the control group was different to that in the 24th hour for the same group. All of these results indicate that both *L. casei* and *L. casei* ΔPyrR can stimulate the polarization of peritoneal macrophages obtained from mice. To detect the cellular compartment in which *L. casei* persists in its interaction with macrophages, we incubated mBMMs with CFDA-SE-labeled bacteria and assessed their colocalization with *L. casei*. Importantly, *L. casei* was deficient in pyrimidine metabolites, which preferentially target peritoneal macrophages in order to efficiently decrease their interaction. Meanwhile, *L. casei* ΔPyrR was observed to be surrounded by macrophages that did not show colocalization with the marker, thereby suggesting that they are able to evade their localization. Therefore, the *L. casei*-associated pyrimidine metabolites modified the internalization and colocalization with macrophages.

## 3. Discussion

Probiotics, strains containing Bifidobacterium and Lactobacillus, are the predominant groups of the gastrointestinal microbiota. Pharmabiotics, postbiotics, and next-generation probiotics are considered as a means to prevent oxidative stress, inflammatory processes, and neurodegenerative and viral diseases through gut microbiome regulation [9]. They have considerable potential for preventive or therapeutic applications in various intestinal diseases. However, most probiotic health claims have not yet been substantiated by experimental evidence via the enzymatic and genetic regulation of different lactic acid bacteria. In addition, the efficacy demonstrated by one given bacterial strain cannot necessarily be transferred to other probiotic organisms. In our study, the interpretation of methods arising from genetics, metabolomics, and proteomics have provided the opportunity to investigate the mechanisms that underlie the action of lactic acid bacteria, such as adhesion to intestinal mucosa, the competitive exclusion of pathogenic microorganisms, and the production of antimicrobial substances. All of these mechanisms should build the interaction of probiotics with the host-associated immune system. Our research involved constructing a PyrR-deficient *Lactobacillus casei* strain that is able to decrease the production of the pyrimidine analog, which further extensively interferes with growth on different pathogenic bacteria.

The five transcriptional units—carB [10], pyrRPBcarA [7], pyrEC, pyrKDbF [11], and pyrDa [12]—for pyrimidine biosynthetic genes are scattered on the chromosome of L. lactis subsp. A pyrR gene encoding a regulatory protein was found to be the first gene of the pyrRPBcarA operon, which has been shown to decrease the expression of all pyrimidine biosynthetic genes by three- to eight-fold in *Lactococcus lactis* [7]. The pyrimidine biosynthetic pathway is initially formed from the formation of carbamoyl phosphate (CP), which also catalyzes the conversion of DHO to orotate with the carAB-PyrB-PyrC-PyrD cassette. Orotic acid, cytosine (C), uracil (U), thymine (T), cytidine (CR), and uridine (UR) are converted to UMP, thereby producing the end-products of pyrimidine in the biosynthetic pathway. Site-directed mutagenesis reduced the UTP and CTP pools that were regulated by transcription attenuation and mediated by a trans-acting repressor in *Lactobacillus plantarum* [13]. The PyrR-type attenuators were found upstream of all the pyrimidine biosynthetic genes, which can be identified in most lactobacilli, except those in *L. casei*. Our study developed the widely applicable and efficient PyrR-based CRISPR-Cas system in *L. casei* for whole genetic deletion, which provided novel insights into the molecular mechanisms of the PyrR gene in relation to the antimicrobial antagonism of metabolic products, competition with pathogens for adhesion, and immunoregulation. The molecular docking results indicated that the pyrR protein attenuates the virulence of biofilm formation and targets the de novo pyrimidine synthesis pathway in S. aureus [14]. To increase the sensitivity of 5-FOA, the double-mutant strain ΔpyrF ΔpyrR in *Geobacillus kaustophilus* was constructed. It was auxotrophic for uracil and resistant to 5-FOA [15]. The parasite PyrR activity was nonspecific and active with respect to uridine, deoxyuridine, pyrimidine, and thymidine.

As a result, the selection (uracil auxotroph) and counterselection (5-FOA) markers can be utilized for developing CRISPR-Cas systems for broad and convenient clinical applications with *L. casei* strains. Additionally, there was no effect on the growth performance in the strains that contained PyrR deletions, which were uracil-auxotrophic hosts. The CRISPR-Cas9-based genome editing system combined with pyrF/5-FOA is both time-saving and efficient with respect to developing gene editing systems for a variety of A. baumannii strains [16]. Moreover, these genetic manipulation methods unveil the drug resistance of oxidative stress-sensing mechanisms within the OxyR gene via the performance of a H_2_O_2_-sensing pocket in an A. baumannii strain. The reduced pyrimidine outside the lactobacillus in mutant bacteria likely reflects differences in PyrR-induced cytotoxicity, which are then exported to the cell surface in the SEM results. In contrast, the disruption of PyrR in the *L. casei* ΔPyrR strain produced severe uracil auxotrophy due to the loss of the de novo pyrimidine synthesis pathway. These findings reveal the importance of ubiquitous surface modifications for PyrR function, which explain the broad lactobacillus colonization and processes that underlie host–pathogen interactions and also highlight the utility of genome-wide CRISPR/Cas9 screenings with respect to explaining their probiotic role.

Today, ineffective first-line antibiotics, emerging pathogenic bacteria, and reduced immunity are becoming critical reasons for the clinical death of patients. Therefore, novel antimicrobial agents from probiotics are urgently needed in the field of drug discovery. These agents have attracted the attention of scientists due to their remarkable biological activity, i.e., their use in anticancer, antiviral, antimicrobial, anti-inflammatory, analgesic, antioxidant, and antimalarial therapies. They are characterized by bacteriocins, hydrogen peroxide, and diacyls, which prevent the proliferation of glucose and inhibit growth [17]. The most famous of these agents are antimicrobial compounds, and pyrimidines, as an integral part of DNA and RNA, play an important role in drug discovery projects with pharmacological and chemical significance [5]. Pyrimidine analogue are classified many heterocycles in antimicrobial drug discovery, and they first attracted our attention due to the remarkable pyrimidine pool in probiotics. The metabolites interact with diverse protein targets in pathogenic microorganisms, leading to the inhibition of FtsZ polymerization, GTPase activity, and bacterial cell division [18]. Furthermore, 5-Fluorouracil was used in a phase III clinical trial as an antibacterial agent [19]. Many Gram-positive and Gram-negative bacteria were found to be susceptible to floxuridine [20]. Additionally, Gram-positive *Streptococci*, *S. aureus*, and *Bacillus* species were more sensitive than Gram-negative bacteria, while the same trend was also found for *L. casei*’s antibacterial activity in our test. An encouraging example of metabolic engineering in Pseudomonas putida resulted in the improvement of microbial bioconversion for ferulic acid consumption via an efficient genome editing strategy [21]. The deletion of a gene that encodes a protein molecule of 1,3-propanediol reductase *in L. reuteri* decreased the production of reuterin when compared to the wild strain, which was identified as a novel antimicrobial variant for killing *E. coli* [22]. Regarding synthetic biology developments, there was an improvement in the regulatory control systems, with the silencing and deletion of the host of lactic acid bacteria metabolites controlling gene expression in vivo at the targeted location with genome editing [23,24]. The programmable autolysis and killing of specific pathogens have been genetically engineered in the bacterial reprogramming of *E. coli* [25,26] and could be adapted to LAB. Most importantly, a court recently ruled that new genome editing methods (including CRISPR-Cas), as ‘non-GMO’ techniques, have recently been greenlit for use in the USA for Cas9-edited plants [27]. LAB could also be considered as microbial cell factories in the development of traditional industrial biotechnologies with green chemical production, which would create an iterative workflow of “Design–Build–Test–Learn”.

The modification of native genes in probiotics could be one approach to enhance the immunomodulation properties of hosts. Although the overall mechanistic understanding of probiotic features in the context of immunoregulation is still mostly lacking, their key role in probiotic metabolites and outer cell surfaces is in the interaction with the probiotic host. Teichoic acids (TAs) stimulate TLRs in mammalian hosts in order to promote dendritic cell differentiation, which subsequently leads to cytokine responses [28] and the D-alanylation of Tas, which decreases the adhesion of *L. reuteri* to host cells [29]. In another study, the deletion of the phosphoglycerol transferase gene was enacted upon the biosynthesis of lipoteichoic acid (LTA) in *Lactobacillus acidophilus*. In addition, the LTA-deficient strain relieved colonic polyposis in a unique mouse model when compared with wild-type *L. acidophilus* [30].

Regarding our study, we conclude that pyrimidine biosynthesis modulates the intracellular nucleotide pools to maintain metabolic homeostasis, thus quenching the host’s innate immunity during biotrophy. The nucleotidyltransferases (CD-NTases) in bacteria were capable of synthesizing the cyclic pyrimidine in bacterial physiology and the modulation of the host’s innate immune system [31]. Moreover, the V. cholerae encoded di-nucleotide cyclase (DncV) works to synthesize the cyclic AMP-GMP molecule, which also plays a beneficial role in the host adaptation and intestinal colonization processes [32]. These pyrimidine adducts are microbial small molecules that are derived from metabolic pathways to activate the mucosal-associated invariant T cell (MAIT) for immunosurveillance [33]. Most importantly, Toxoplasma gondii KU80 has been used to delete the orotidine-5′-monophosphate decarboxylase (OMPDC) and uridine phosphorylase (UP) genes to obtain an avirulent non-reverting pyrimidine auxotroph strain, providing novel tools for the dissection of the host Th1 immune response with respect to infection [34]. Our study also analyzed the functional capacity of the pyrimidine pathways by deleting the PyrR salvage activity in *Lactobacillus casei*, which controls the direct access to the host cell pyrimidine biosynthetic pathway with or without pyrimidine interacting with the host’s immune cells, which indicates that pyrimidine stimulated peritoneal macrophage polarization in the mice.

Overall, the examples we have provided illustrate how, for food-grade bacteria in general and probiotic lactobacilli to be identified as novel variants, genetic engineering approaches work to promote human health. As long as the application of CRISPR-based technologies continues to expand the genetic toolboxes for LAB, we believe that LAB will have the potential to modulate microbiota and host immunity. In particular, the genome editing of LAB will be much more effective in terms of eradicating the target microbes in new specific strains. Advancements in this field must be accompanied by the consideration of matters and topics such as the ongoing CRISPR intellectual property battles, the enabling of technological advances in LAB, biological safety, policies and regulations, and consumer acceptance. Nonetheless, the ongoing microbiology renaissance, fueled by CRISPR technological advances and our increasing awareness of LAB, remains strong.

## 4. Materials and Methods

### 4.1. Bacterial Strains and Plasmids

The *Lactobacillus casei* was separated from Min pig, which was cultured in MRS broth (Sigma, St. Louis, MO, USA) without shaking at 37 °C. The antibiotic concentration was 10 mg per liter with erythromycin in MRS media. Competent cells of *E. coli* TG-1 were used as the cloning host. All *E. coli* harboring pCLNICK series plasmids grew on the LB media with kanamycin (50 mg/L) at 32 °C. The *Lactobacillus casei* gene-editing plasmid pLCNICK was kindly supplied by Yang Sheng (Key Laboratory of Synthetic Biology, CAS, Shanghai, China). All the bacterial strains and plasmids for cloning, as well as the *L. casei* and *L. casei* ΔPyrR mutants used in this study, are listed in Table 2.

### 4.2. Plasmid Construction of PyrR-Deficient L. casei by CRISPR-Cas9^D10A^

All of the steps of gene cloning and plasmid construction are shown in Figure 7. According to the general principles of primer design, the PyrR gene and up/down homology arms were cloned with the up-stream/down-stream primer. The pyrimidine regulatory gene (PyrR) was cloned as the target gene from *Lactobacillus casei*. The PyrR and the upstream/downstream homology arms (HASup and HASdown) were amplified via PCR. The PCR product (HASup-PyrR-HASdown) was connected to the cloning vector of pMD19, which was used as the template to clone the HASup and HASdown genes. Both the HASup and HASdown genes were cloned into the cloning vector. The cloning HASup and HASdown genes were connected via fusion PCR, the PCR product of which was cloned into pMD19. To construct pMD19-sgRNA-HASup-HASdown for genome deletion, the sgRNA was connected to a single fragment of HASup-HASdown via overlap extension PCR. The resulting cassette of sgRNA-HASup-HASdown with XbaI and ApaI restriction endonuclease sites was ligated to a cloning vector of pMD19. To generate the genome-editing plasmid, which was named pLCNICK-PyrR-HAS-sgRNA, the backbone of the P23 promoter for Cas9^D10A^ (Em^r^), the promoter of Pldh for sgRNA (sgRNA), and Has (obtained via the restriction endonuclease of XbaI and ApaI from the pMD19-sgRNA-HASup-HASdown plasmid) were used. All of these recombinant plasmids were chemically transformed into competent cells of *E. coli TG1*.

### 4.3. Transformation of Lactobacillus casei for Electroporation

The constructed plasmids were transformed into *L. casei* via electroporation. Competent cells of *L. casei* were prepared according to the following protocol: In total, 2 mL of *L. casei* was cultured into a 100 mL MRS+ 2% glycine media and incubated overnight at 37 °C until the optical density in the logarithmic phase was reached without shaking at 5000 rpm and at 4 °C. The *L. casei* cells were chilled on ice for 10 min and harvested via centrifugation at 5000 rpm and 4 °C for 10 min, washed twice with 40 mL of ice-cold EPWB (NaH_2_PO_4_ 0.6 mmol/L, MgCl_2_ 0.1 mmol/L), and then washed once in 40 mL of ice-cold EPB (NaH_2_PO_4_ 0.6 mmol/L, MgCl_2_ 0.1 mmol/L, 0.3 mol/L Sucrose). The *L. casei* cells were centrifuged for 10 min at 5000 rpm and 4 °C. The iced *L. casei* competent cells were resuspended with 1 mL of EPB, which was used prior in electroporation.

Competent cells of *L. casei* were mixed with 1 μg of the plasmid pLCNICK-PyrR-HAS-sgRNA and subsequently transferred into a precooled 2 mm electroporation cuvette at 2.2 kV, 200 Ω, and 25 μF with a Bio-Rad GenePulser Xcell. The precooled MRS+ 500 mM sucrose broth (1 mL) was immediately added to the electroporation cuvette, which was then transferred to a culture tube to recover growth for 5 h at 37 °C without shaking. The recovered cells were grown on MRS plates containing erythromycin (5 μg/mL) for 4 or 5 days, and single colonies were randomly selected for growth in MRS broth containing erythromycin (5 μg/mL). All colonies were used to extract the genome as the template determining the PyrR was knocked out using the PCR method. The *L. casei* mutation strain with the PyrR gene deletion in its genome was constructed to identify the positive clone (named *L. casei* ΔPyrR). All of the PCR products obtained from identifying the positive clone were sequenced to confirm the deleting strain.

### 4.4. Prokaryotic Expression of Recombinant Plasmid and Its Anti-Mice PyrR Protein Polyclonal Antibody Preparation

The recombinant protein of PyrR was expressed by the expressing vector of Cold Shock Expression System pCold™ DNA (Takara Bio Inc., Kusatsu, Shiga, Japan), and the recombinant proteins were induced via IPTG and purified using the His-tag Protein Purification Kit (Beyotime, Shanghai, China). The purified protein was immunized in mice to produce an anti-mice PyrR protein polyclonal antibody, which was taken from the serum of mice that was used in the Western Blot analysis part of our study.

### 4.5. Batch Fermentation, Growth Curve, and Lactic Acid Production

Fermentation was carried out in 100 mL of MRS media in 250 mL Erlenmeyer flasks. The *L. casei* and *L. casei* ΔPyrR strains were statically cultivated in a MRS medium at an initial density OD600 of 0.05 (MRS Broth, Becton, Dickinson and Company, New York, NY, USA). The cells of the cultures were incubated at 37 °C without shaking. For time course experiments, the samples were taken every 2 h for an analysis of the bacterial population density.

### 4.6. Determination of 5-FOA Susceptibility

The strains *L. casei* and *L. casei* ΔPyrR were cultivated on solid MRS plates that consisted of 5-FOA in amounts of 250 μmol/L. The *L. casei* and *L. casei* ΔPyrR strains were incubated on MRS plates at 37 °C for 3–4 days. The growth and morphology of the colonies that were incubated on the plates were evaluated to assess the tolerance of the *L. casei* and *L. casei* ΔPyrR strains to 5-FOA.

### 4.7. Fermentation and Detection of Metabolic Production

Fermentation was carried out with 10 mL of MRS media in 250 mL Erlenmeyer flasks. The *L. casei* and *L. casei* ΔPyrR strains were statically cultivated in MRS media at an initial density of OD600 of 0.05. The cells of the cultures were incubated at 37 °C without shaking for 24 h. The suspensions and cells of *L. casei* and *L. casei* ΔPyrR were quantified via high-performance liquid chromatography (HPLC) to study metabolic production. The samples were obtained through using the following conditions: All samples and precooled buffers (20% H_2_O, 40% methanol, 40% acetonitrile *v*/*v*) were mixed together via ultrasonication in an ice bath for 30 min. The separations were centrifuged for 30 min at 12,000 rpm and at 4 °C. All of the samples were vacuum-dried and subsequently resolved using a 100 μL solution (50% H_2_O, 50% acetonitrile *v*/*v*).

The raw data from the LC-MS data were directly preprocessed using the Progenesis QI software 2.0 to facilitate an automatic search of the selected databases. Data from the positive and negative modes were mixed and processed together for a further multivariate statistical analysis that included sample names, peak intensity, retention time (RT), compound molecular weight, and metabolite identification.

### 4.8. Antibacterial Activity Assay

To investigate whether the *L. casei* and *L. casei* ΔPyrR strains could induce the damage and growth of pathogenic microorganisms, their antimicrobial activity was evaluated using the Oxford cup antibacterial experiment to determine the changes in the bacteriostatic activity of metabolic production and the lactobacillus body composition from the *L. casei* and *L. casei* ΔPyrR strains. *Bacillus subtilis*, *Salmonella typhimurium*, and *Staphylococcus aureus* were used as the indicator bacteria in the in vitro experiment. The indicator bacteria suspension was diluted to 10^8^ cfu/mL, and 0.2 mL of each diluted bacterial culture suspension was inoculated on LB plates and evenly spread using a coater. Immediately afterwards, the Oxford cup method was performed by creating four vertical holes on the surface of the solidified LB medium, and 0.1 mL of each sample was added to each hole with PBS (negative control), antibiotics (positive control), a microbial supernatant, and a lactobacillus culture media composition in each plate of both the *L. casei* and *L. casei* ΔPyrR strains. Then, the Petri dishes were incubated at 37 °C for 10 h, and the diameter of the circular antibacterial zone was measured to determine the antagonistic activity of the microbial supernatant and lactobacillus body composition. The experiment was repeated three times under the same conditions to avoid the occurrence of errors.

### 4.9. Scanning Electron Microscopy Analysis of the Genome Editing of Lactobacillus casei

The morphological comparison of the genome editing of the *Lactobacillus casei* that deleted PyrR was further validated by microscopic analyses via the use of a scanning electron microscope (SEM) (Hitachi Limited, Tokyo, Japan). The control and deleting strains were cultured in an optical density at 37 °C without shaking. The samples were centrifuged and rinsed with 1× phosphate-buffered saline (PBS), fixed in a 2.5% glutaraldehyde solution that was buffered with 1× PBS, and incubated overnight. The next step was to dehydrate the samples with increasing concentrations of 20, 40, 60, 80, and 100% ethanol and then incubate them in 100% ethanol overnight at 4 °C. The dried samples were mounted on SEM stubs that were coated with a gold sputter for observation using a Quanta™ 250 FEG SEM (Thermo Fischer Scientific, Waltham, MA, USA). The SEM images were captured using PCI imaging software (Quartz Imaging Corp., Vancouver, BC, USA).

### 4.10. Fluorescent Tracking of Lactobacillus in Direct Contact with Macrophages

Briefly, the recombinant lactobacilli were grown in liquid MRS at 37 °C and washed three times with phosphate-buffered saline (PBS). The *L. casei* and *L. casei* ΔPyrR strains were stained with carboxyfluorescein diacetate and succinimidyl ester (CFDA-SE) for 30 min in 1 mL of CFDA buffer at 37 °C on a vertical rotator in a dark environment. To stain all of the cells, the lactobacilli were washed twice with 5 mL of PBS in order to remove excess CFDA-SE. The stained *L. casei* and *L. casei* ΔPyrR strains were resuspended into a single-cell suspension with 1 mL of a 1640-cell culture media. The peritoneal macrophages were collected as previously described [35]. The mice were injected into the peritoneum with 5 mL of the 1640-cell culture media for peritoneal macrophage collection. The mice’s abdomens were gently rubbed for 2 min to enable the release of the peritoneal macrophages into the culture media from the abdominal cavity. The peritoneal fluid was sucked out and transferred into a tube, which was subsequently centrifuged for 5 min at 1000 r/min. Then, the macrophages were resuspended with the 1640-cell culture media, which contained 10% fetal calf serum, penicillin, and streptomycin, at 37 °C under a 5% CO_2_ air atmosphere overnight.

To explore whether *L. casei* may promote the polarization of peritoneal macrophages via the pyrimidine production of the PyrR gene, we established the *L. casei* and *L. casei* ΔPyrR strains, as well as a peritoneal macrophage co-culture system, which was separately extracted from the mice. The CFDA-SE-labeled *L. casei* and *L. casei* ΔPyrR strains were contained inside the collected macrophages and cultured overnight in 12-well plates. After incubation at 37 °C under a 5% CO_2_ air atmosphere for 7 h, the plates were washed two times with pro-cold PBS and fixed with 4% paraformaldehyde for 30 min at RT. The wells were washed with pro-cold PBS and blocked with 1% BSA for 2 h at RT. F4/80^+^ served as a macrophage marker, which was used to measure the activity of the peritoneal macrophages with respect to the stimulation of pyrimidine production from the *L. casei* PyrR gene. The peritoneal macrophages were blocked via incubation with anti-F4/80 antibody binding for 30 min at RT in a dark environment. Then, the peritoneal macrophages were washed and stained with DAPI for 10 min, and images of them were captured via fluorescent microscopy (Leica Microsystems, Madrid, Spain).

### 4.11. Data Analysis

The metabolomics data were analyzed using different modules of the web-based platform MetaboAnalyst (https://www.metaboanalyst.ca, accessed on 24 October 2022). The data were normalized using logarithmic transformation and auto-scaling. Fold change analysis and *t*-tests were conducted to determine the fold change and significance of each identified metabolite. This process was performed in triplicate, and all of the groups were examined according to their independent measurements to adequately support the statistics and results derived from our experiments. The average recovery and colonization times over time were compared between groups using the repeated-measures analysis of variance model with Bonferroni’s correction using SPSS 26 software (IBM). Statistically significant effects (*p* < 0.05) were further analyzed, and mean values were compared using Tukey’s significant difference test.

## 5. Conclusions

Taken together, our findings illustrate that we successfully constructed an auxotroph for PyrR-deficient *L. casei*. The evidence presented in this study shows that the PyrR gene is an active and key component in the regulation of pyrimidine biosynthesis within the metabolic profiles of a *L. casei* supernatant. Furthermore, this supernatant inhibited the growth of a wide range of pathogens in vitro. The complementary regulation of PyrR-deficient *L. casei* could not inhibit the growth of pathogens nor enable immune regulation in macrophages collected from mice. Thus, the deletion of PyrR in *L. casei* resulted in the loss of the probiotic characteristics of antimicrobial antagonism and immunoregulation.

## Figures and Tables

**Figure 1 microorganisms-11-02371-f001:**
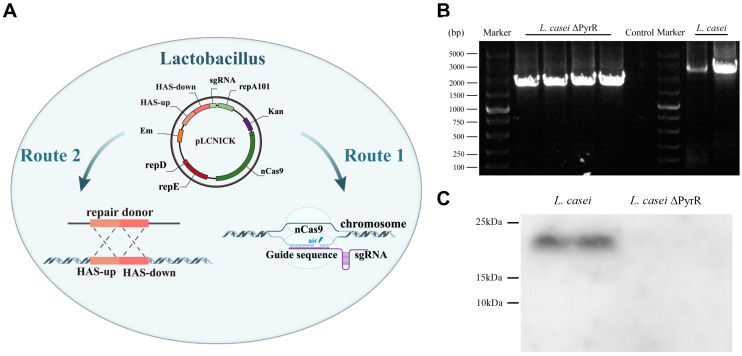
Chromosomal deletion of PyrR gene. (**A**) Schematic diagram of the two steps of pLCNICK-mediated PyrR knockout: the precise cutting PyrR gene in the homologous arm gene editing sites and the deletion pasting of PyrR DNA. (**B**) Colony PCR assays were conducted using testing primers. (**C**) The protein expression of PyrR were analyzed via Western blotting, which showed a wild strain of *L. casei* to be an immunoreactive band and the *L. casei* Δ*PyrR* strain to be a non-immunoreactive band.

**Figure 2 microorganisms-11-02371-f002:**
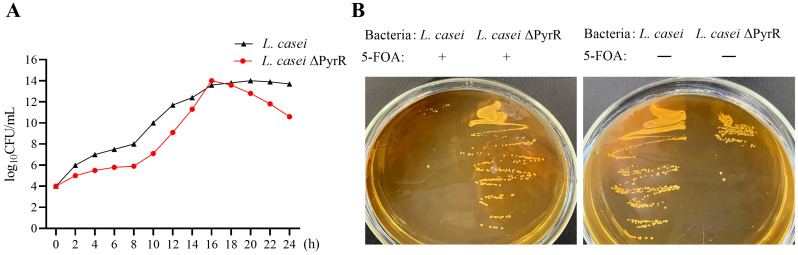
Characterization of growth in the *L. casei* and *L. casei* ΔPyrR strains. (**A**) The growth curves of *L. casei* and *L. casei* ΔPyrR strains grow in the MRS media. The curves were calculated via Standard Plate Count (SPC) every 2 h, respectively. (**B**) To measure 5-FOA resistance, the wild-type and mutant cells were grown on MRS plates with 10 mM 5-FOA and incubated for 5 days.

**Figure 3 microorganisms-11-02371-f003:**
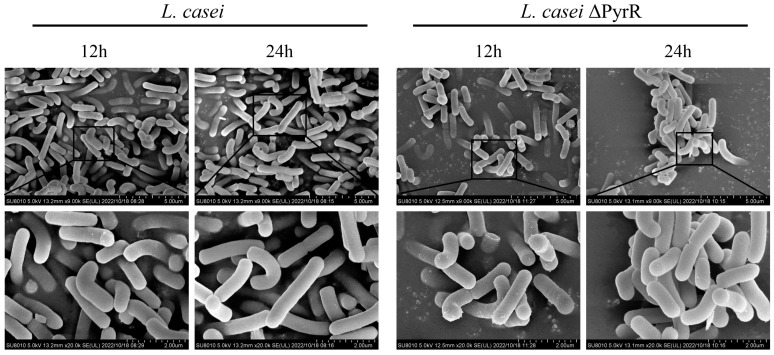
Morphological characteristics of the genome editing of *Lactobacillus casei*. The *L. casei* and *L. casei* ΔPyrR were observed by using a scanning electron microscope. The images of the samples were captured at the 12th hour and 24th hour of the growth stage.

**Figure 4 microorganisms-11-02371-f004:**
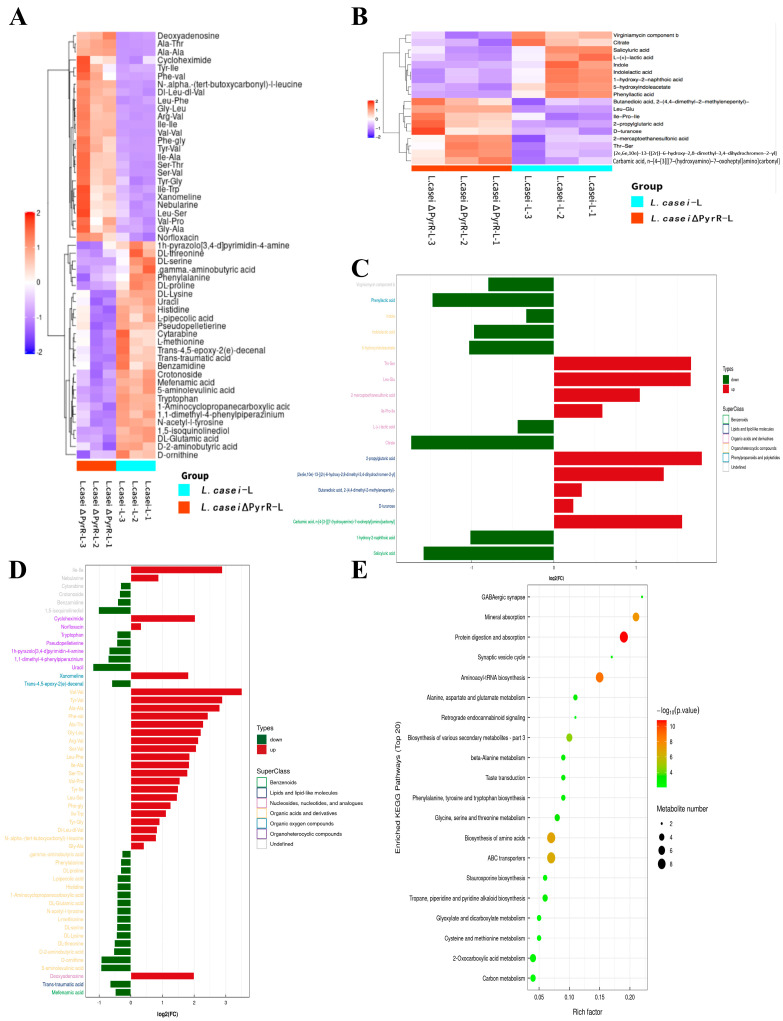
Global, non-targeted metabolomes of *L. casei* and *L. casei* ΔPyrR cell-free supernatants. (**A**) Heat map of 53 metabolites ranked according to the magnitude of fold differences between the *L. casei* and *L. casei* ΔPyrR group within the positive-ion mode. (**B**) Heat map of 18 metabolites ranked according to the magnitude of fold differences within the negative-ion mode. (**C**) Major differential metabolites were classified into pathways associated with biological processes from the negative-ion mode. (**D**) Major differential metabolites were classified into pathways associated with biological processes from the positive-ion mode. (**E**) Bubble map of metabolic pathway enrichment analysis. The X-axis enrichment factor (RichFactor) is the value of the differential metabolites annotated in the pathway. Larger values indicate a greater proportion of differential metabolites annotated in the pathway. The dot size represents the number of differential metabolites annotated in the pathway.

**Figure 5 microorganisms-11-02371-f005:**
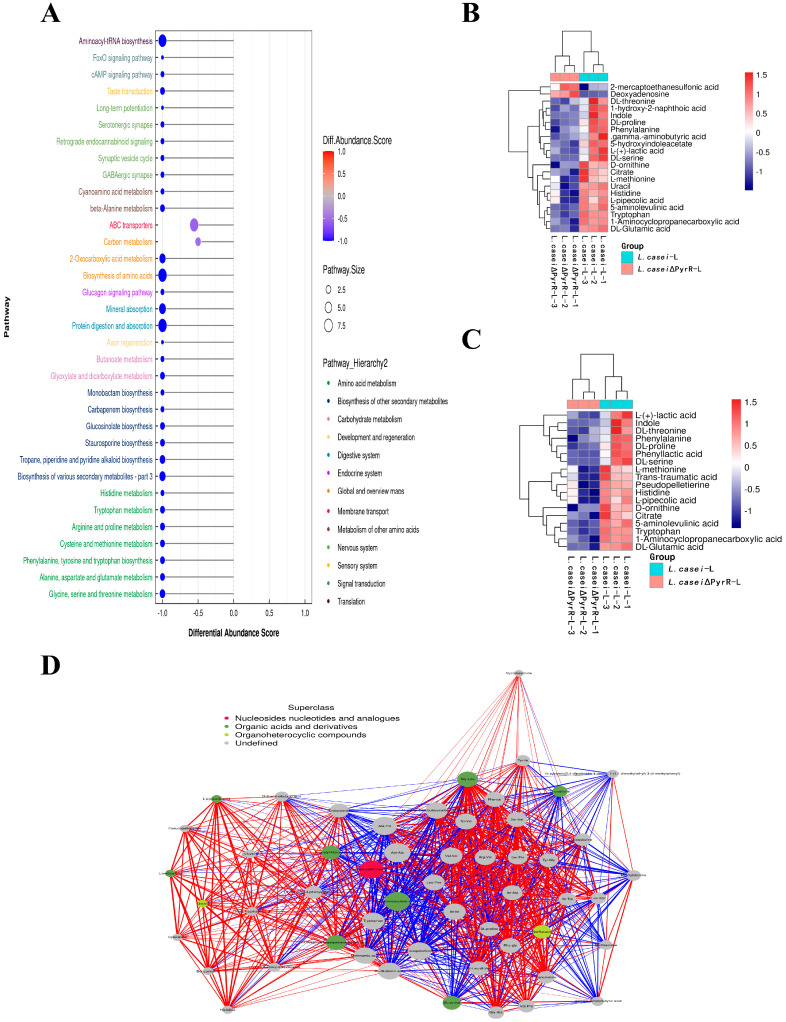
Significantly changed pathway in metabolite set enrichment analysis with PyrR editing. (**A**) Metabolite set enrichment analysis showing metabolic pathways enriched in the *L. casei* ΔPyrR compared to the *L. casei*. (**B**) The expression of differential metabolites in the pyrimidine and purine pathway (shown via a heatmap). (**C**) The expression of differential metabolites in the biosynthesis of secondary metabolites pathway (shown via a heatmap). (**D**) Correlation network analysis based on the integrated metabolomic and differential abundance score.

**Figure 6 microorganisms-11-02371-f006:**
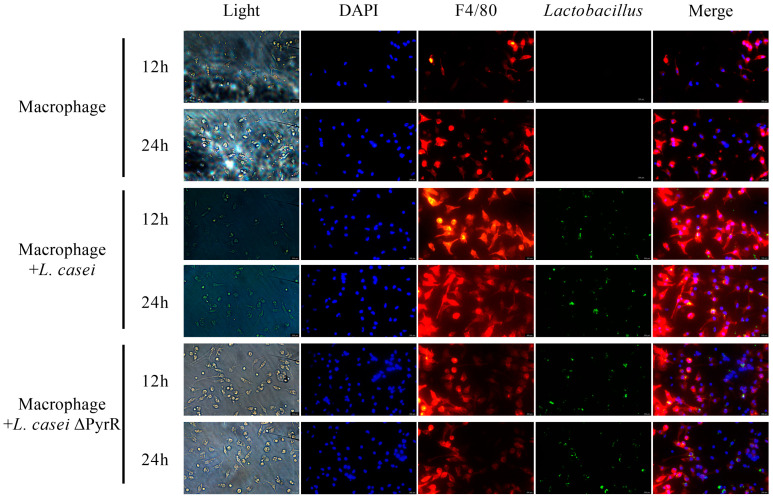
Representative *L. casei* and *L. casei* ΔPyrR (green), F4/80 (red), and DAPI (blue) immunofluorescence staining of peritoneal macrophages from mice (Scale bar, 500 μM). The histograms indicate the percentages of *L. casei* and *L. casei* ΔPyrR (green) or positive cells in macrophages (red).

**Figure 7 microorganisms-11-02371-f007:**
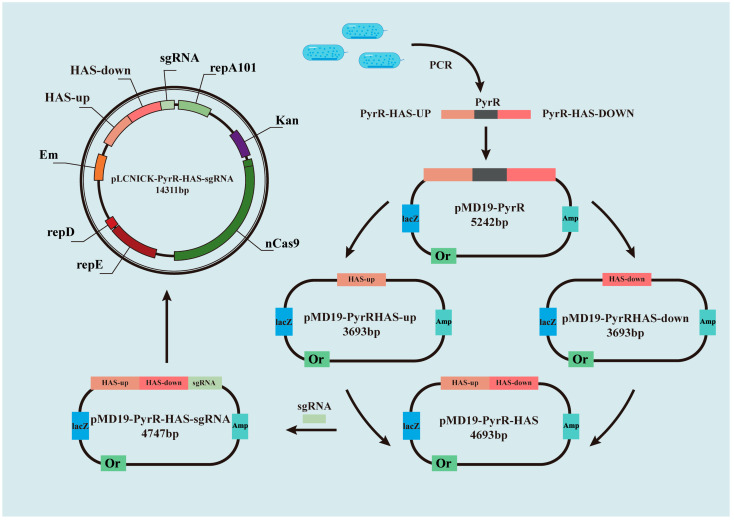
Schematic diagram of pLCNICK-mediated *L. casei* PyrR gene knockout.

**Table 1 microorganisms-11-02371-t001:** Plasmids and strains used in this study.

Strain or Plasmid	Relevant Characteristics
Stains	
*L. casei* LC2W	Wild type
*E. coli* DH5α	Commercial host for cloning
TG1	Commercial competent cell for cloning
BL21	Commercial competent cell for prokaryotic expression
Plasmids	
pLCNICK	repA101(Ts), kan, P23-Cas9^D10A^,P_ldh_-sgRNA, Has(PyrR), repD, repE and Em^r^
pPROHTA	Iac1, MCS, Amp^r^, ori, f1 ori
pMD19	ColE1 ori, LacZ operator, Amp^r^
pMD19-PyrR	ColE1 ori, LacZ operator, Amp^r^, PyrR
pPROHTA-PyrR	Iac1, MCS, Amp^r^, ori, f1 ori, PyrR
PyrR-Has-updown	PyrR, HAS-up(PyrR), HAS-down(PyrR)
pMD19-PyrRHAS-up	ColE1 ori, LacZ operator, Amp^r^, HAS-up(PyrR)
pMD19-PyrRHAS-down	ColE1 ori, LacZ operator, Amp^r^, HAS-down(PyrR)
pMD19-PyrR-HAS	ColE1 ori, LacZ operator, Amp^r^, HAS(PyrR)
pMD19-PyrR-HAS-sgRNA	ColE1 ori, LacZ operator, Amp^r^, HAS(PyrR), sgRNA
pLCNICK-PyrR-HAS-sgRNA	repA101(Ts), kan, P_23_-Cas9^D10A^,P_ldh_-sgRNA, Has(PyrR), repD, repE and Em^r^, sgRNA

**Table 2 microorganisms-11-02371-t002:** Oligonucleotides used in this study.

Oligonucleotide	Sequence (5′ → 3′)
arm-pyrR-merge	GTTGTGAAATGCCATCTGATGTGCCACCCT
arm-pyrR-left-down	TGATGTGCCACCCTTTCTTAT
arm-pyrR-right-up	GATGGCATTTCACAACGACGA
pyrR106-up	GGGCCCCTCAATTCCTGAAGCCAAGC
pyrR106-down	TCTAGACCGCGGGTTTTGATGCCGACTAAACCGATGAAGGCAAAC
pyrR9-up	GGGCCCCTCAATTCCTGAAGCCAA
pyrR9-down	TCTAGACCTCGTCTACAACTTGTTTTGACACCGATGAAGGCAAACA
pyrR-exup	GAATTCATGGCACAGTCAAAACAAGT
pyrR-exdown	CTCGAGTTAGCCCTCCATCCTTTCG
arm-pyrR-yz-up	CTCAATTCCTGAAGCCAAGC
arm-pyrR-yz-down	ACCGATGAAGGCAAACAAGATGG
arm-pyrR-merge-up	TAAGAAAGGGTGGCACATCAGATGGCATTTCACAACGACGAAG

**Table 3 microorganisms-11-02371-t003:** Bacteriostatic diagram of deleting the PyrR gene in *L. casei*.

*Bacillus subtilis*	*Lactobacillus* Culture Suspension	
	200 µL	100 µL
Oxford cup diffusion method	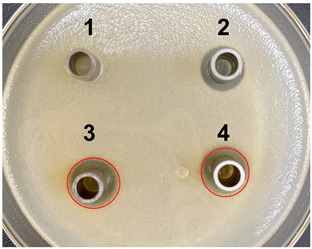	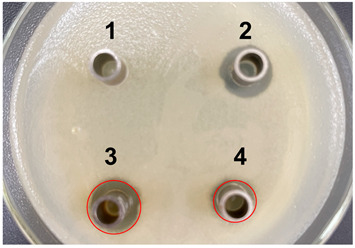
Growth inhibition diameter	3: 13.56 mm 4: 12.04 mm	3: 13.40 mm 4: 10.24 mm
* **Staphylococcus aureus** *	***Lactobacillus* Culture Suspension**	
	**200 µL**	**100 µL**
Oxford cup diffusion method	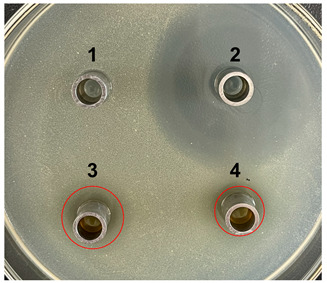	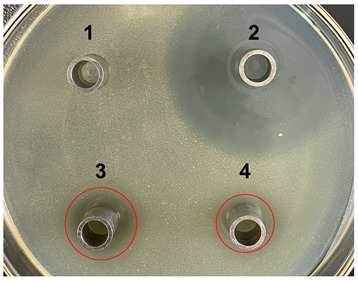
Growth inhibition diameter	3: 12.36 mm 4: 11.16 mm	3: 12.32 mm 4: 9.64 mm
* **Salmonella** *	***Lactobacillus* Culture Suspension**	
	**200 µL**	**100 µL**
Oxford cup diffusion method	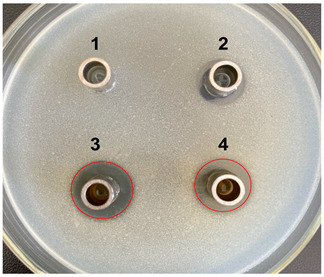	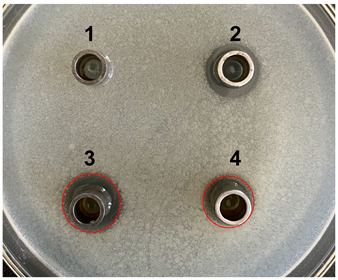
Growth inhibition diameter	3: 17.92 mm 4: 16.16 mm	3: 17.00 mm 4: 14.02 mm

The outer diameter of the Oxford cup is 8 mm. The liquids in the plate were as follows: PBS buffer of negative control group (No. 1), antibiotic of positive control group (No. 2), suspension from *L. casei* (No. 3), and suspension from *L. casei* ΔPyrR (No. 4).

## Data Availability

The raw data supporting the conclusions of this article will be made available by the corresponding authors without undue reservation.

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
