# Peer review of "Loss in the Antibacterial Ability of a PyrR Gene Regulating Pyrimidine Biosynthesis after Using CRISPR/Cas9-Mediated Knockout for Metabolic Engineering in *Lactobacillus casei"

_microorganisms, 2023, doi:10.3390/microorganisms11102371_

Round 1

Reviewer 1 Report

In this manuscript, the authors study A Gene that Regulates the Pyrimidine Biosynthesis and is important in Antibacterial Ability for Metabolic Engineering in Lactobacillus casei. In this study, a PyrR-deficient L. casei strain was constructed using the CRISPR-Cas9D10A tool. There were some changes in its basic biological characterization, such as its growth curve, auxotroph, and morphological damage. The metabolic profiles of the supernatant between the PyrR-deficient and wild strains revealed the regulation of the synthesis of genetic material and of certain targeting pathways and metabolites. In addition, the characteristics of the PyrR-deficient strain were significantly altered as it lost the ability to inhibit the growth of pathogens. Authors identified PyrR-regulating pyrimidine biosynthesis, which further improved its internalization and colocalization with macrophages. Evidence shows that the PyrR gene is a key active component in L. casei supernatant for the regulation of pyrimidine biosynthesis against a wide range of pathogens.

The manuscript is very well written and covers all aspects scientifically. Data is solid and I found no major loopholes in this study. It can be published in its current form.

Minor

Author Response

Thank you for your comments. And we appreciate your recognition for our study. Our research was the first paper with the deleting gene and functional research in lactobacillus to indicate that the pyrimidine biosynthesis, which could affect growth curve, auxotroph, morphological damage and antibacterial ability of Lactobacillus. Our research provides the new strategy in the probiotic, which will much more precise to find the function in the postbiotic. However, the potential mechanism how the pyrimidine metabolites allow inhibit the growth of pathogens was not be indicated in the study. We think that it is a worthy in-depth study in the future, and we will explore the detailed mechanism on the antibacterial ability of pyrimidine metabolites in the following study.

Reviewer 2 Report

Reviewer comments

Manuscript: microorganisms-2595051 - A PyrR Gene Regulating Pyrimidine Biosynthesis Lost its Antibacterial Ability Using CRISPR/Cas9-Mediated Knock-out for Metabolic Engineering in Lactobacillus casei

The authors studied a PyrR gene regulating pyrimidine biosynthesis lost its antibacterial ability using CRISPR/Cas9-mediated knock-out for metabolic engineering in Lactobacillus casei. The authors identified PyrR-regulating pyrimidine biosynthesis, which further improved its internalization and colocalization with macrophages. Evidence shows that the PyrR gene is a key active component in L. casei supernatant for the regulation of pyrimidine biosynthesis against a wide range of pathogens.

The data analysis methods are correct.

The English of the text is well written and well readable but needs additional checking with a professional translator.

The uniqueness of the text is more than 90% by AntiPlagiarism.NET.

The text contains some misspellings and typos. Also need to expand the part of the discussion.

There are some comments and questions:

1) Line 234, 242, 245 - Bacillus subtili - should be - Bacillus subtilis.

2) Line 260 - Staphyloccocus - should be - Staphylococcus.

3) In the Table 2. - explain all groups 1, 2, 3, 4.

4) Line 300 - after the sentence - Probiotics, i.e., strains containing Bifidobacterium and Lactobacillus, are the predominant groups of the gastrointestinal microbiota. - add citation (Danilenko et al., 2021) - add to the References - Danilenko, V.N.; Devyatkin, A.V.; Marsova, M.V.; Shibilova, M.U.; Ilyasov, R.A.; Shmyrev, V.I. Common inflammatory mechanisms in COVID-19 and Parkinson’s diseases: the role of microbiome, pharmabiotics and postbiotics in their prevention. Journal of Inflammation Research 2021, 14, 6349–6381. doi: 10.2147/JIR.S333887

5) How does the PyrR gene allow inhibit the growth of pathogens?

6) Figure 4. - the text very small. Please increase it.

7) In the sentence -  The Bacillus subtilis, Salmonella typhimurium and Staphylococcus aureus were used the indicator bacteria in the vitro experiment. - W

8)Why the authors studied - Why used as the indicator bacteria.

9) 

Please improve the manuscript according to the above comments.

The English of the text is well written and well readable but needs additional checking with a professional translator.

Author Response

Comment 1, Line 234, 242, 245 - Bacillus subtili - should be - Bacillus subtilis.

Thanks so much for your useful suggestion. We have corrected the name by reviewer’s advice. All the correcting part have been marked with red in the manuscript.

Comment 2, Line 260 - Staphyloccocus - should be - Staphylococcus.

Thank you for pointing out this problem in manuscript. We have revised it and have corrected it with red in the manuscript.

Comment 3, In the Table 2. - explain all groups 1, 2, 3, 4.

We used Bacillus subtilis, Staphylococcus aureus and Salmonella as the indicator bacteria in the Table 2 to detect the antibacterial activities of the L. casei DPyrR. The liquid in the four groups were the same, which is divided into 1) PBS buffer as the negative control group, 2) antibiotic as the positive control group, 3) suspension of culture media from L. casei, 4) suspension of culture media from L. casei DPyrR. All these details were added at the bottom of table 2, which have been marked with red.

Comment 4, Line 300 - after the sentence - Probiotics, i.e., strains containing Bifidobacterium and Lactobacillus, are the predominant groups of the gastrointestinal microbiota. - add citation (Danilenko et al., 2021) - add to the References - Danilenko, V.N.; Devyatkin, A.V.; Marsova, M.V.; Shibilova, M.U.; Ilyasov, R.A.; Shmyrev, V.I. Common inflammatory mechanisms in COVID-19 and Parkinson’s diseases: the role of microbiome, pharmabiotics and postbiotics in their prevention. Journal of Inflammation Research 2021, 14, 6349–6381. doi: 10.2147/JIR.S333887

Thanks so much for your useful suggestion. It is a comment that worth modification to the manuscript. This reference provides a new view for the study of probiotic in the genome edition. The genome editing probotic could be used in different diseases. We have added it to the References.

Comment 5, How does the PyrR gene allow inhibit the growth of pathogens?

In the study, the antibacterial activities have been a lower sensitivity in the deleting PyrR gene group compared with the wild strain of Lactobacillus casei. Therefore, the result could indicate that the PyrR gene has antibacterial activity with lower level of PyrR-regulating pyrimidine biosynthesis.

Comment 6, Figure 4. - the text very small. Please increase it.

Thanks for your comment about the Figure 4. Due to the original size of the text is very small, we deeply regret that we could not increase it as possible as we can. But we can provide all the original picture for you.

Comment 7, In the sentence - The Bacillus subtilis, Salmonella typhimurium and Staphylococcus aureus were used the indicator bacteria in the vitro experiment. Why the authors studied - Why used as the indicator bacteria.

We want to choose some typical bacteria in the vitro experiment of antibacterial activity. The Bacillus subtilis, Salmonella typhimurium and Staphylococcus aureus are the common probiotic and pathogenic bacteria. All these probiotic and pathogenic bacteria contained gram-negative and gram-positive bacteria, bacillus and coccus. Therefore, the indicator bacteria were used to evidence antibacterial activity that the PyrR gene is a key active component in L. casei supernatant for the regulation of pyrimidine biosynthesis, which play a wide range of inhibiting growth for pathogens.

Comment 8, Please improve the manuscript according to the above comments.

Thank you for all your professional comments. We have improved the manuscript seriously according to the comments. We will accept all the suggestions from you for our manuscript.